# Point-of-care ultrasound (POCUS): Assessing patient satisfaction and socioemotional benefits in the hospital setting

**Evan A. Balmuth**[1]*, **Danny Luan**[1], **Deanna Jannat-Khah**[2,3], **Arthur Evans**[1], **Tanping Wong**[1], **David A. Scales**[1]

1 Department of Medicine, Division of General Internal Medicine, NewYork-Presbyterian Hospital/Weill Cornell Medical College, New York, New York, United States of America, 2 Department of Medicine, Division of Rheumatology, NewYork-Presbyterian Hospital/Weill Cornell Medical College, New York, New York, United States of America, 3 Department of Medicine, Hospital for Special Surgery, New York, New York, United States of America

☝ These authors contributed equally to this work.
* eab4002@med.cornell.edu

**Data Availability Statement:** All relevant de-identified data are within the paper and its Supporting Information files.

## Abstract

Point-of-care ultrasound (POCUS) is an imaging modality used to make expedient patient care decisions at bedside. Though its diagnostic utility has been extensively described, POCUS is not yet considered standard of care in inpatient settings. Data from emergency department settings suggest that POCUS may yield socioemotional benefits beyond its diagnostic utility; furthermore, elements of the POCUS experience are known to promote placebo effects. These elements likely contribute to a placebo-like "POCUS positive care effect" (PPCE) with socioemotional benefits for receptive patients. Our objective is to provide the first characterization of the PPCE and its facilitating factors in an inpatient setting. In this novel mixed-methods study, we recruited 30 adult patients admitted to internal medicine floors in an urban academic medical center, recorded observations during their routine POCUS encounters, and administered post-encounter surveys. We conducted complementary quantitative and qualitative analyses to define and assess the magnitude of the PPCE. We also aimed to identify factors associated with and facilitating receptiveness to the PPCE. The results indicated that POCUS improves patients' satisfaction with their hospital providers and care overall, as well as perceived care efficiency. Mutual engagement, strong therapeutic alliances, and interpreting POCUS images to provide reassurance are most closely associated with this PPCE. Patients who have lower anxiety levels, less severe illness, and received efficient care delivery during their hospitalizations are most receptive to the PPCE. We conclude that diagnostic POCUS has the potential to exert a positive care effect for hospitalized patients. This PPCE is associated with modifiable factors at the patient, provider, and environment levels. Together, our findings lay the groundwork for an optimized "therapeutic POCUS" that yields maximal socioemotional benefits for receptive patients.

**Funding:** This research was funded by the Weill Cornell General Internal Medicine Primary Care Innovation Seed Grant (https://careinnovation.weill.cornell.edu/rfps/seed-grants-innovative-primary-care-proposals-2021-2022), awarded to DAS and TW. The funders did not play any role in the study design, data collection and analysis, decision to publish, or preparation of the manuscript.

**Competing interests:** The authors have declared that no competing interests exist.

## 1. Introduction

Point-of-care ultrasound (POCUS) is an imaging modality performed at bedside to make expedient care decisions [1, 2]. First used in the emergency department (ED), POCUS has been increasingly leveraged by inpatient internal medicine providers. Beyond its established diagnostic advantages, studies of POCUS in the ED point toward additional benefits including improved patient satisfaction with care [3], confidence in the POCUS provider [4], and health-care-associated cost savings [5, 6]. However, the socioemotional benefits of POCUS are only beginning to be explored and have not yet been characterized in the inpatient setting. Furthermore, the contextual factors which facilitate these benefits remain undefined.

The POCUS experience involves several aspects that have been previously shown to mediate placebo effects and modulate emotional states [7]. These include several features shared with the physical exam such as touch, spending time with the provider, and active listening [8, 9]; the use of sophisticated instruments, devices, and procedures [10]; provider communication and displays of empathy [11–13]; and sharing ultrasound images [14]. However, POCUS is technically not a placebo; whereas true placebos mimic a specific therapy while lacking its key therapeutic element [15, 16], POCUS is an inert yet complex intervention firmly intertwined with care processes in the hospital [7]. Recognizing crucial roles for care contexts and therapeutic processes, the term "placebo effect" has been increasingly replaced with "positive care effect" to describe therapeutic benefits resulting from clinical encounters [17, 18]. Therefore, "POCUS positive care effect" (PPCE) may be a more apt term for patients' socioemotional responses to POCUS.

Novel frameworks to understand placebo effects' complex psychosocial mechanisms have begun to emerge in recent years. One compelling framework proposes that social and contextual factors, such as the patient-provider relationship and the physical treatment environment, influence patients' expectations and mindsets; in turn, certain mindsets and expectations will be most conducive to a placebo effect [19, 20]. The patient's mindset in particular seems to have an instrumental role in shaping both objective and subjective indicators of wellbeing [20–24]. Therefore, an investigation of psychosocial dynamics and patient mindsets in the POCUS encounter will be instrumental for understanding the PPCE.

Ultimately, a greater appreciation for this therapeutic potential may facilitate more widespread POCUS utilization—yielding substantial diagnostic and socioemotional benefits for hospitalized patients. Although the evidence basis for its diagnostic utility has been firmly established, POCUS is not yet considered standard of care in the inpatient setting. Furthermore, despite the increasing use of POCUS and recognition of its diagnostic advantages on inpatient wards, its propensity to yield socioemotional benefits for hospitalized patients remains poorly understood. With thorough characterization of the PPCE, POCUS can be optimized to elicit maximal socioemotional benefits for patients.

Therefore, in this mixed-methods study, we seek to characterize the PPCE in the inpatient setting. Specifically, we define the PPCE and its facilitating factors through surveys of patient satisfaction paired with qualitative POCUS observations. Our novel design leverages complementary quantitative and qualitative approaches to provide a uniquely thorough illustration of the PPCE, and ultimately generate hypotheses for future investigation.

## 2. Methods

### 2.1 Study design, participants, & sampling

This was a mixed-methods cross-sectional study based on qualitative analysis of POCUS encounter field notes and quantitative analysis of post-encounter surveys. Thirty adult patients

admitted to internal medicine floors at NewYork-Presbyterian Hospital/Weill Cornell (NYP/WC) were identified by convenience sampling for POCUS observations, audio recordings, and surveys. Patients were eligible if they were age 18 years or older, admitted with a chief complaint of chest pain or shortness of breath, and would receive POCUS as part of their routine care. Patients were ineligible if they were pregnant, breastfeeding, unable or unwilling to independently provide consent. Potential participants were identified through chart review by one of two trained observers (DL and EB), medical students trained in POCUS for hospital internal medicine who were not involved in the patients' clinical care. Chart review screening and recruitment were completed between September 21st and November 15th, 2021. At bedside, the observers requested patients' verbal consent for audio recording of their POCUS encounters and surveys. The observers emphasized that they were not part of the care team and that all survey responses would remain anonymous. Observers and providers communicated through a phone interpreter with patients who preferred languages other than English. Patient recruitment and data analysis were conducted in tandem, and recruitment continued until the conclusion of qualitative analysis (see section 2.3.2).

The study was conducted according to the guidelines of the Declaration of Helsinki, and approved by the Institutional Review Board (IRB) of Weill Cornell Medicine (protocol number 19–10020908). Verbal consent for audio recording of POCUS encounters and surveys was obtained from all participants involved in the study. Given the routine nature of the observed care, and to minimize influences of observation, informed consent for participation in the study beyond audio recording was not requested; this was in accordance with a waiver of informed consent provided by the IRB of Weill Cornell Medicine.

## 2.2 Data collection

**2.2.1 Demographic & clinical characteristics.** Patients' pertinent demographic and clinical characteristics (Table 1) were collected by chart review and—as with all data in this study—recorded directly into REDCap, a secure web platform and data management system [25, 26]. The authors retained access to information that could identify individual participants after initial eligibility screening in order to complete chart review of demographic and clinical characteristics. All information that could identify individual participants was stored securely in REDCap [25, 26].

**2.2.2 POCUS encounter observations & surveys.** After obtaining verbal consent for audio recording, observers accompanied POCUS providers (including internal medicine attendings, physician assistants (PAs), and POCUS fellows) for the patients' routine POCUS encounters in their hospital rooms or in the ED. (Though all patients had been admitted, some POCUS encounters occurred in the ED when inpatient beds were not yet available.) Providers were briefed on the purpose of the study and instructed to perform POCUS as they normally would for routine care (e.g., to assess for fluid collections, volume status, cardiac function, etc.). Providers were also instructed to ignore observers in the room and avoid referencing the research study to minimize potential influences of observation. The observers re-entered the patients' rooms with the providers, began audio recording, and observed the encounters while recording electronic field notes into REDCap [25, 26]. POCUS was performed using the GE Venue Go Ultrasound System. Field notes focused on verbal and non-verbal observations of patient-provider interactions and the immediate care environment. Video recordings were not obtained as many encounters occurred in places where complete privacy was not possible (e.g., in the ED). After the POCUS encounter, observers returned to complete a survey consisting of a mixture of Likert-style and open-ended questions (S1 Table). These questions were adapted from surveys utilized in prior studies to assess patients' experiences [3, 4, 12]. Post-

**Table 1. Demographic and clinical characteristics of study participants who completed post-encounter surveys ($n$ = 27).**

| Variables | n | Median (IQR) | % |
|---|---|---|---|
| Age[1] | | 63.0 (59.0–75.0) | |
| Sex | | | |
| Male | 16 | | 59.3 |
| Female | 11 | | 40.7 |
| Preferred Language | | | |
| English | 23 | | 85.2 |
| Other[2] | 4 | | 14.8 |
| Chief Complaint | | | |
| Chest pain | 10 | | 37.0 |
| Dyspnea | 9 | | 33.3 |
| Chest pain + dyspnea | 8 | | 29.6 |
| Chief complaint severity[1,3] | | 2.0 (1.0–4.0) | |
| CCI[1,4] | | 5.0 (2.0–6.0) | |
| Patient-rated magnitude of PPCE[1,5] | | 5.0 (4.3–5.0) | |
| Patient-rated POCUS provider empathy[1,6] | | 7.0 (6.0–7.0) | |

[1]Categorical variables are presented as sample size (n) and column percentage (%); continuous and ordinal (Likert-style) variables are presented as median (interquartile range). Superscripts are added to the continuous and ordinal variables.

[2]Other languages included Spanish, Creole, Bengali, and Romanian (one patient each).

[3]Chief complaint severity was assessed using a 1–5 Likert scale, with 5 being the most severe and 1 being the least severe.

[4]Charlson Comorbidity Index (CCI) is an aggregate score with points assigned for age and 16 comorbidity categories, validated for predicting mortality [27, 28]; the maximum possible score is 37.

[5]Magnitude of POCUS positive care effect (PPCE) was calculated as the average of responses to 3 survey questions which utilized a 1–5 Likert scale, with 5 being 'strongly agree' and 1 being 'strongly disagree' (see S1 Table).

[6]Provider empathy was calculated as the average of all 3 responses to the empathy survey which utilized a 1–7 Likert scale, with 7 being 'strongly agree' and 1 being 'strongly disagree' (see S1 Table).

encounter anxiety level was assessed with the six-item short-form State-Trait Anxiety Inventory (STAI-6) [29, 30]. Patients and POCUS providers were asked to estimate the encounter duration and rate their rapport in the encounter. Quality of rapport between patient and provider was also assessed and recorded by the observer, as were contextual factors like the amounts of patient-provider interaction and chaos during the POCUS encounter.

## 2.3 Data analysis

**2.3.1 Quantitative data & analysis.** Quantitative data were analyzed with R [31] and GraphPad Prism 9, which was also used to generate scatterplots [32]. Descriptive statistics were calculated as median and interquartile range (IQR) for continuous and ordinal variables; counts and percentages were calculated for categorical variables. Magnitude of PPCE was calculated as the average of three Likert-style patient ratings for the extent to which POCUS improved patients' interactions with providers, care efficiency, and satisfaction with care. POCUS provider empathy was calculated as the average of three Likert-style patient ratings for the extent to which their provider understood their perspective, emotions, and concerns. PPCE was tested for correlations with demographic and survey response data through 26 independent nonparametric Spearman tests and two point-biserial tests (for categorical variables),

which were not adjusted for multiple comparisons (see section 4.5). Actual duration of encounter was correlated with patient- and provider-estimated duration through Spearman tests; linear regressions with a null hypothesis of slope = 1 were subsequently applied to both correlations. The difference in time (Δ Time) was calculated by subtracting actual duration from patient- or provider-estimated duration of encounter. Charlson Comorbidity Index (CCI) was calculated for each patient following chart review [27, 28]. STAI-6 score was calculated by summing the anxiety-present features and the inverse of the anxiety-absent features [29, 30]. STAI-6 score was correlated with Δ Time (patient) using a Spearman test.

**2.3.2 Qualitative data & analysis.** Audio recordings were manually transcribed and combined with corresponding field notes into a text document imported into NVivo 12 Plus [33]. All 30 transcripts were systematically and iteratively coded through thematic analysis [34]. Two analysts (DL and EB) were responsible for coding all 30 transcripts. The first five transcripts were coded jointly, employing a discursive approach until consensus was achieved on a preliminary codebook. Initial deductive codes were defined based on features of POCUS encounters that were hypothesized to contribute to a PPCE; these initial codes were then iteratively modified and expanded through an inductive coding approach informed by grounded theory [35]. After establishing the preliminary codebook, all subsequent transcripts were coded separately. Analysts met regularly with additional research team members (DS and TW) to discuss emerging themes and discursively resolve coding discrepancies. As additional codes emerged, analysts returned to previously coded transcripts to identify any instances of the new codes. Coding continued until theoretical saturation, the point at which no new themes emerged from analyzing additional transcripts [35–37]. Throughout this process, the analysts reflected upon and discussed their positionality—specifically, how their clinical backgrounds in patient care and POCUS shaped their perceptions and data interpretations.

In addition to thematic analysis, the analysts leveraged observational field notes to analyze the extent to which each POCUS interaction was influenced in more or less positive ways by different factors. For example, patients appearing comfortable, relaxed, and smiling were considered reflections of a positive experience. In contrast, patients appearing uncomfortable, anxious, or experiencing symptom exacerbation were consistent with a less positive experience. Patients sometimes provided verbal information consistent with a positive experience, such as expressing amazement during the encounter or describing positive emotions during post-encounter surveys. In contrast, complaints or expressions of pain were considered consistent with a less positive experience. Analysts reflected upon and discussed the entirety of observed verbal and non-verbal evidence to determine the relative influence of different observed factors on positivity. This process of coding at the level of the overall interaction did not follow any previously-established method in thematic analysis; however, it paralleled aspects of interactional sociolinguistics and conversation analysis—approaches to discourse analysis which consider language and social context to understand human interactions [38]. These approaches have been previously utilized to study successfulness of clinical encounters, physician communication, and clinical expertise [39–42].

## 3. Results

### 3.1 Quantitative results

**3.1.1 Baseline patient characteristics.** Thirty patients consented to participate in the study, among whom 27 completed post-encounter surveys. Baseline demographic and clinical characteristics of the 27 patients who completed post-encounter surveys are shown in Table 1. Patients' median age was 63.0 years (IQR 59.0–75.0); 59% were men and 41% women. The majority of patients (85%) were English-speaking. The median patient-reported post-

encounter chief complaint severity was 2.0 out of 5 (IQR 1.0–4.0). Patients' median CCI was 5.0 out of 37 (IQR 2.0–6.0).

**3.1.2 Survey-derived composite scores.** The median patient-rated magnitude of PPCE—a composite score representing the average of three Likert-style patient ratings on a 1–5 scale for the extent to which POCUS improved patients' interactions with providers, care efficiency, and satisfaction with care—was 5.0 out of 5 (IQR 4.3–5.0) (Table 1). Sixteen patients (59%) provided the maximum composite score of 5.0 out of 5. The lowest composite score among all patients was 1.7 out of 5. The median patient-rated POCUS provider empathy value—a composite score representing the average of three Likert-style patient ratings for the extent to which providers understood patients' perspectives, emotions, and concerns—was 7.0 out of 7 (IQR 6.0–7.0) (Table 1). Fourteen patients (52%) provided the maximum composite score of 7.0 out of 7. The lowest composite score among all patients was 3.0 out of 7. The individual post-encounter survey questions and response medians are listed in S1 Table. Notably, patients' median post-encounter STAI-6 score was 9.0 on a scale from 6 to 24 (IQR 7.0–14.0).

**3.1.3 Correlating variables with PPCE.** Among the 28 variables tested for correlation (Spearman rho, $r_s$) with PPCE, the following had a significant positive correlation: patient-rated POCUS provider empathy ($r_s = 0.67$; $p = 0.0001$), patient-rated confidence in POCUS provider ($r_s = 0.52$; $p = 0.006$), observer-rated rapport in POCUS encounter ($r_s = 0.46$; $p = 0.02$), and provider-rated rapport in POCUS encounter ($r_s = 0.41$; $p = 0.04$). Higher post-encounter STAI-6 scores (indicating higher anxiety) were negatively correlated with PPCE ($r_s = -0.49$; $p = 0.009$) (Table 2).

**Table 2. Correlations with magnitude of PPCE.**

|  | R value[1] | | p value |
|---|---|---|---|
| **Assessments of POCUS encounter** | | | |
| Patient-rated POCUS provider empathy | + 0.67 | | 0.0001 |
| Patient-rated confidence in POCUS provider | + 0.52 | | 0.006 |
| Observer-rated rapport in POCUS encounter[2] | + 0.46 | | 0.02 |
| Provider-rated rapport in POCUS encounter[3] | + 0.41 | | 0.04 |
| **Assessments of healthcare experience** | | | |
| General satisfaction with providers' skills and abilities | + 0.70 | | < 0.0001 |
| General quality of interactions with providers | + 0.56 | | 0.002 |
| Quality of symptom management | | + 0.51 | 0.007 |
| Overall satisfaction with care in this hospital | + 0.44 | | 0.02 |
| General satisfaction with providers' knowledge | + 0.43 | | 0.02 |
| Overall satisfaction with hospitalization | + 0.43 | | 0.03 |
| **Patient characteristics** | | | |
| Post-encounter anxiety level[4] | − 0.49 | | 0.009 |
| Patient age | − 0.27 | | 0.17 |
| Patient sex | + 0.13 | | 0.51 |
| Patient-rated severity of chest pain or dyspnea (chief complaint) | + 0.04 | | 0.83 |

[1]Spearman $r_s$ was calculated for all variables except sex, for which point-biserial $r_{pb}$ was calculated.

[2]One pair of data points was excluded from this analysis due to missing observer ratings.

[3]Two pairs of data points were excluded from this analysis due to missing provider ratings.

[4]Anxiety level was assessed with the STAI-6, a six-item short-form of the STAI (State-Trait Anxiety Inventory). The aggregate score calculated by summing the anxiety-present features and the inverse of the anxiety-absent features. The survey utilized a 1–4 Likert scale, with 1 being "not at all" and 4 being "very much." The maximum possible score is 24 and the minimum possible score is 6 [29, 30].

The PPCE also had positive correlations with factors beyond the POCUS encounter, including patient-rated general satisfaction with all providers' skills and abilities ($r_s = 0.70$; $p < 0.0001$), all providers' knowledge ($r_s = 0.43$; $p = 0.02$), quality of interactions with all providers ($r_s = 0.56$; $p = 0.002$), quality of symptom management ($r_s = 0.51$; $p = 0.007$), overall care in this hospital ($r_s = 0.44$; $p = 0.02$), and the overall hospitalization ($r_s = 0.43$; $p = 0.03$) (Table 2). Of note, these ratings encompassed global perceptions of all the patients' hospital providers (e.g., attendings, residents, PAs, nurses, etc.) and care from the ED to the inpatient floor.

There was no significant association between PPCE and patient age, patient sex, or patient-rated chief complaint severity (Table 2). (See S2 Table for all tests performed.)

**3.1.4 Time estimations.** The correlation between patient-estimated and actual durations of the POCUS encounters differed significantly from the line of identity (S1 Fig). Provider-estimated durations closely approximated actual durations, and the correlation between provider-estimated and actual durations did not differ significantly from the line of identity (S1 Fig). Twenty-five patients (93%) underestimated their encounter durations. The median actual duration was 17.0 minutes (IQR 9.0–25.0); in contrast, the median patient-estimated duration was 10.0 minutes (IQR 7.5–20.0), a difference of 7.0 minutes or 41% underestimate when comparing overall median values. The median difference between patient-estimated and actual durations (Δ Time)—a value calculated for each individual POCUS encounter—was -5.0 minutes (IQR -10.0– -2.0) (S1 Table). The largest underestimates were among patients with lower anxiety scores (S2 Fig).

## 3.2 Qualitative results

**3.2.1 Factors contributing to a positive POCUS experience.** *3.2.1.1 Mutual engagement driven by curiosity & education.* Mutual engagement was a defining feature of many positive encounters. Providers consistently met their patients' curiosity with education—proactively explaining images and fielding questions in a real time back-and-forth while both intently watched the ultrasound monitor:

*He explained everything he was doing. Everything he did he just explained it to me. He was just very nice, and you understood the way he explained to me how the blood flows, and my liver. No one ever said that to me before, anything about my liver, and he did.*

Many patients were pleased to learn something new about their body:

*It was a really great experience with her because she did the ultrasound and she was teaching me what was going on. She taught me about the left side and right side. She taught me a couple things about my heart that I didn't know.*

Several patients eagerly described the images that had been explained to them: "*it was good, I got to see my heartbeat.*" Notably, mutual engagement in each of these encounters occurred despite elements of chaos in the care environment—loud dialysis machines, voices intruding from the hallway, lack of privacy from another patient in the room—which could have detracted from a positive experience.

*3.2.1.2 Therapeutic alliance defined by respect & trust.* A trusting, respectful therapeutic alliance appeared to be characteristic of most positive encounters. Strong alliances became most apparent during potentially unpleasant parts of the POCUS experience. A recurring example occurred when providers empathetically apologized for pressing with the ultrasound probe or reaching over the patient; in many cases, the patients' responses revealed an underlying

understanding and trust that these were necessary parts of the job: *"the provider says, 'pardon my reach.' The patient says, 'you got to do what you got to do.' The patient has her eyes open looking straight ahead and appears to be comfortable."* In some cases, patients gave permission to proceed with potentially uncomfortable probing: *"'so that's the vein in your heart. You see it?' The patient nodded. 'Alright, sorry for pressing so hard.' The patient said, 'if you need to press harder, go ahead.'"* Such interactions reflected providers' respect for their patients' comfort, as well as patients' respect for their providers' POCUS abilities and perception that their task is important. In a subset of encounters, satisfied patients held an unwavering trust in their providers from the moment they walked into the room: *"[the ultrasound doctor] came to do his job and I need to trust. In my experience, I believe I have to accept the care for my own good, for my own health."* Notably, this Spanish-speaking patient reported trust despite low English proficiency and a provider with low Spanish proficiency who did not use an interpreter.

*3.2.1.3 Powerful & reassuring machines.* Many patients who had a positive POCUS experience were impressed by the power and capabilities of ultrasound technology. Some who had prior experiences with ultrasound were amazed to realize the extent of anatomy that it could be used to investigate—and appreciated how thoroughly their providers leveraged its full capabilities: *"what I remember most about the ultrasound was that, most of the time when you get an ultrasound, they're doing your kidney, your heart, but he did everything, your neck, every organ. He looked at everything."* Among ultrasound's vast applications, its diagnostic utility was the focus of many patients who were most satisfied with their POCUS experience. When patients learned that their providers had used POCUS to rule out pathology, they were left feeling positive and more at ease: *"it was good. No pain, no misery, no nothing. As long as the results were good. Once he said there's nothing life-threatening, I'm not worried at all."* In some instances, along with a sense of relief, POCUS confirmation of good organ function provided a sense of validation for years of health promotion: *"the patient asks again, 'so far, the heart looks good?' The provider says, 'yes, the valves and the chambers look good.' The patient says, 'thank God, I used to be so active in the gym."* Notably, in both of these instances, the patients were observed to be feeling positive overall when certain pathologies were ruled out—despite the identification of other potentially concerning findings (fluid around the heart and around the lungs, respectively).

**3.2.2 Factors detracting from a positive POCUS experience.** *3.2.2.1 Severe illness & pathological findings.* Especially severe illness during the POCUS encounter tended to detract from a positive experience. In some instances, as for this patient with severe shortness of breath during an asthma exacerbation, maneuvers intended to facilitate effective POCUS ended up aggravating symptoms:

*The patient sat looking somewhat dejected, looking downward or straight ahead. Dr. X supported the patient by holding his right arm. "Sir, are you doing okay?" Dr. X asked. He said yes, then said, "my head is spinning." "Could you take a deep breath for me, sir?" The patient inspired deeply, then began coughing. The patient was breathing rapidly.*

In many instances, pathological findings in the setting of severe illness provoked concern—as this patient with a possible myocardial infarction expressed through anxious questions:

*The doctor said, "it looks like you may have some fluid in your lungs." The patient asked, "are you sure?" The doctor then explained some ultrasound signs that suggest the presence of fluid. "What do you do if there's fluid? Is there any way to get rid of it?" The doctor replied, "by fixing the heart. . . also some medications can help." The patient exhaled deeply.*

Some patients, such as one woman with metastatic pancreatic cancer and a large fluid collection around her lung, expressed their concern directly through exclamations and body language:

*The provider explains the images and says, "there's considerable amount of fluid that has to come out." The patient says, "oh my God, that's a lot of fluid." The provider says, "that has to come out and you'll feel a whole lot better." The patient looks forward at the ultrasound screen. The provider shows her the heart below the lungs. The patient looks at her feet, at the ground, and at the wall. The provider stands right behind the patient and explains what she sees as she looks at the lung. The patient looks concerned but the provider provides reassurance.*

*3.2.2.2 Chaotic care environment with delays & lack of privacy*. A more chaotic environment —with disruptions by passersby and neighbors—also tended to detract from a positive POCUS experience. Notably, while such chaotic environments are typical of most urban hospitals, the degree of chaos can vary depending on numerous factors such as whether the patient is in the ED or in a single or shared inpatient room. Some patients explicitly noted concerns about privacy during their exams:

*The patient was sitting up in bed. The PA moved the divider that separated him from the patient adjacent to him, and he said, "this isn't privacy." The PA said, "this is what privacy looks like in the ED." She then drew the curtains. The patient joked, "I'll have to bring my Cadillac in here."*

Others appeared distracted by neighboring patients and clinical teams, and did not engage actively in their own encounters. For example, throughout her POCUS encounter one patient paid close attention to a neighboring clinical interview and exam, watching as the other team exited and re-entered her shared room. The patient appeared disinterested in her ongoing POCUS exam, and no communication occurred with her POCUS provider. Aside from chaotic hospital rooms and ED spaces, several patients were more preoccupied with chaotic care delivery—especially regarding treatment delays: *"prior to starting the ultrasound exam, the PA entered to introduce herself. The patient asked why she hadn't received her blood pressure pills and seemed upset."* In another instance, frustration with care delays similarly impeded rapport-building early in the encounter:

*The doctor entered to identify the patient, and then asked if she could look at the patient's heart with the ultrasound. The patient said, "you don't know why I'm here—I'm here because of the clogged artery," and then he asked when they would do the cardiac catheterization.*

## 4. Discussion

### 4.1 Predominance of PPCE & satisfaction

The magnitude of PPCE was rated highly by all but one participant, as POCUS improved patients' satisfaction with their providers and overall care, in addition to perceived care efficiency. Our finding that patients were generally very satisfied with POCUS mirrors that of prior publications, across ED and outpatient settings [3, 4, 43–45]. To our knowledge, this is the first study to assess patient satisfaction with POCUS in an inpatient internal medicine setting.

Our qualitative data from field notes and interviews illuminated nuances in patient satisfaction that were not as readily detectable in Likert-style ratings: notably, thematic analysis revealed a substantially less positive encounter overall for the only patient who provided a low PPCE rating (1.7 out of 5), suggesting that qualitative observations may mirror quantitative scores. In addition, the three patients who did not complete post-encounter surveys (and were therefore not included in subsequent analyses) spoke primary languages other than English; prior work suggests that cultural values likely impact individuals' interactions with authority, which could have played into these patients' decisions to participate in the study but not complete the surveys [46].

Importantly, patient age and sex did not corelate with magnitude of PPCE.

## 4.2 Factors associated with greater PPCE & promoting positive POCUS experiences

Two factors that correlated strongly with PPCE were patient-rated POCUS provider empathy and confidence in their POCUS provider. Our qualitative analyses similarly revealed the importance of a respectful and trusting therapeutic alliance. Trust, perceived competence, and empathy are known to be core characteristics of patient-provider relationships that shape mindsets receptive to placebo effects [20]. Notably, the effect not only makes patients feel better, but in some cases appears to make them physiologically better and more confident in promoting their own health; patient-rated provider empathy has been previously associated with quicker time to recovery from cold viruses [11], better glycemic control [12], and increased patient empowerment to manage their illnesses [13].

Relatedly, observer and provider ratings of rapport in the POCUS encounter independently correlated with the PPCE. From a qualitative standpoint, mutual engagement driven by curiosity and education represented the elements of rapport most conducive to a positive POCUS experience. Along these lines, previous studies have shown that both verbal and non-verbal components of patient-provider communication not only modulate subjective patient outcomes like satisfaction, confidence, anxiety level, and pain [47–52], but also impact objective outcomes such as length of stay and arrhythmias after cardiac surgery [53, 54]. Provider education has also been shown to increase reassurance for patients with acute low back pain, likely by shifting cognitive appraisal of their condition [55]. Together, these data suggest that the PPCE is closely tied to the quality of the patient-provider relationship—especially its elements of empathy, trust, and mutual engagement through communication.

Interestingly, patient-rated rapport, as opposed to provider- and observer-rated rapport, did not correlate with the PPCE. This is likely explained by a ceiling effect, with patients rating rapport very highly across the board, whereas provider and observer ratings were more variable. Also notably, though mutual engagement was identified as a key factor in the qualitative data, observer-rated degree of interaction did not correlate with PPCE. While observers rated the degree of interaction highly as long as *either* the patient or provider was verbally engaged, our qualitative data emphasized the importance of *mutual engagement* and bi-directional interaction as a core facilitator of the PPCE.

Our qualitative data additionally revealed powerful technology, synergizing with providers' expert reassurance via image interpretation, as key drivers of a positive POCUS experience. Patients' overall positive outlooks seemed to result from improved understandings of pathological processes, as providers' ultrasound interpretations shed light on previously frightening unknowns rather than pure reassurance. By combining direct ultrasound visualization with the provider's immediate interpretation, reassurance (when appropriate), and treatment plan, patients were able to quickly understand which organs were healthy, which were causing their

symptoms, how bad it was, and what could be done about it. Sophisticated diagnostic procedures and the reassurance they may provide have previously been shown to exert strong placebo effects in unblinded trials. Two trials utilizing cardiac catheterization for stable angina and stenting for those who met criteria demonstrated significant improvements in chest pain —even for those who did not receive stents [56]. This "faith healing" effect may be related to similar improvements in pain resulting from sham surgeries [56–58]. Provider interpretation is also instrumental in shaping patient mindsets receptive to placebo effects [20]. Physicians' clear communication about diagnoses, such that patients understand and agree about the nature of their conditions, has been shown to accelerate recovery from illness in primary care settings [59]. Relatedly, providers' ability to relieve "fears of the unknown" by interpreting fetal ultrasound images has been shown to reduce maternal anxiety [47]. To our knowledge, this is the first study to identify a role for provider reassurance in promoting patient satisfaction with POCUS encounters.

Quantitative data also revealed that patients' satisfaction with their hospital care and providers beyond the POCUS encounter correlated with the PPCE. Other contextual factors identified in our qualitative data mediated the PPCE, especially elements detracting from the experience (see section 4.3). Predictably, patients' various hospital interactions and care experiences appear to influence their expectations and mindsets going into the POCUS encounter [20].

## 4.3 Factors associated with lesser PPCE & detracting from positive POCUS experiences

Our quantitative data revealed an inverse correlation between post-encounter anxiety level and PPCE. This finding aligned with our qualitative data, which indicated that severe illness and pathological ultrasound findings—both of which could increase anxiety levels [60, 61]— detracted from a positive POCUS experience. Furthermore, prior research has shown that heightened anxiety levels increase the risk of feeling dissatisfied with information received during hospitalization, the friendliness of healthcare personnel, and the hospital setting overall [62]. Importantly, our study focused on "state anxiety," or temporary anxious emotional states, rather than clinical anxiety disorders.

Though we observed severe illness and pathology detracting from positive POCUS experiences, patients' symptom severity ratings did not correlate with PPCE. Whereas observers' perceptions of disease severity and pathology were influenced by numerous pieces of information, including patient appearance, physical signs, medical history, and an ability to directly interpret ultrasound images, these may not have overlapped with patients' subjective experiences of symptoms alone. Moreover, how POCUS modulated patients' symptoms is unclear without pre- and post-encounter surveys. Nonetheless, we may infer that higher anxiety levels, more severe illness, and pathological findings could shift patients out of mindsets that would be amenable to the PPCE.

Observed contextual factors, specifically treatment delays and lack of privacy, also detracted from a positive POCUS experience. These findings echo prior work implicating both privacy and perceived time waiting for care as important contributors to patient satisfaction in the ED [63, 64]. Such factors likely influence patients' mindsets and contribute to negative care expectations [20], making them less receptive to the PPCE. Surprisingly, observer-rated environment chaos did not correlate with PPCE in our quantitative analysis. Though observers' chaos ratings captured noticeable features of the immediate environment (e.g., noisy monitors and interruptions), qualitative data pointed to factors like privacy and treatment delays that may be overarching issues throughout an entire hospitalization.

### 4.4 POCUS influencing time perceptions

Most patients perceived a shorter-than-actual duration of their POCUS encounter; notably, patients who perceived a larger time discrepancy tended to have lower anxiety levels and higher PPCE ratings. However, providers did not demonstrate a discrepancy between actual and estimated encounter durations. Therefore, it appears that time seems to "fly by" faster for patients as the magnitude of the PPCE increases and anxiety level decreases. In contrast to our findings, a limited body of work suggests that longer perceived time with providers is associated with greater patient satisfaction [65, 66]. However, the relationship between time estimations and satisfaction can be complicated by patients' emotional states, which are also known to modulate time perceptions [67]. Ultimately, the relationship between patient satisfaction, emotional state, and time perception likely depends on the clinical scenario. To the best of our knowledge, no study has evaluated how perceived time relates to satisfaction with procedures such as POCUS, and future studies will be needed to further explore this topic.

### 4.5 Limitations

First, this descriptive cross-sectional study only observed and surveyed patients during and after POCUS. This approach was not designed to illustrate a pre- versus post-POCUS effect, but rather to characterize the elements and contributors to more positive or less positive POCUS experiences. With these elements intrinsically intertwined in hospital POCUS experiences, it is challenging to distinguish the relative contributions of each element at the patient, provider, and environment levels. Nonetheless, asking patients to rate their POCUS experience allowed us to quantitatively define a positive care effect. Future studies with pre- as well as post-POCUS surveys and examining potential extensions of the PPCE to symptom improvement will be needed. In addition, providers' behavior could have been influenced by observer bias [68]. Although we instructed providers to perform POCUS as they normally would for routine care, ignore observers in the room, and avoid referencing the research study, observer bias could be further mitigated in future studies by asking patients to evaluate their POCUS encounters without direct observation. Furthermore, our study in a single, urban academic center utilizing convenience sampling has limited generalizability. Additional work is needed to address linguistic and cultural differences surrounding conceptions of pain, satisfaction, and care expectations. Future studies should implement non-Likert style surveys to improve sensitivity and response variability [69]. In addition, we conducted 28 independent correlation tests; however, with $\alpha$ set to 0.05, we would anticipate only one to two false positive results. Therefore, commonly-applied corrections such as Bonferroni were deemed overly strict, and we have elected to acknowledge the slightly elevated risk of statistical false positives [70] which does not substantially detract from our hypothesis-generating goals for this study.

## 5. Conclusions

In this study, we applied a novel mixed-methods approach to provide the first characterization of the PPCE and its facilitating factors in the inpatient hospital setting. Based on our findings, we hypothesize that POCUS can be optimized to elicit a maximal PPCE for receptive patients, in a form we recommend calling "therapeutic POCUS." Specifically, in the right context, the PPCE comprises socioemotional benefits including increased satisfaction with providers and overall care, as well as perceived care efficiency. Encounters that maximize mutual engagement, therapeutic alliance, and reassurance through image interpretation are likely most conducive to therapeutic POCUS; patients who have lower anxiety levels, less severe illness, and have experienced efficient care delivery during their hospitalization may be most receptive to

therapeutic POCUS. These encounter and patient characteristics, intrinsically intertwined as components of the hospital POCUS experience, can be understood to shape patient mindsets most receptive to the PPCE. Future studies implementing pre- and post-POCUS assessments, randomization, and post-discharge follow-up will be best positioned to test and further characterize therapeutic POCUS. However, we caution that indiscriminately utilizing POCUS to yield a PPCE carries important risks. For example, surreptitiously using a diagnostic tool for therapeutic effect could be detrimental to trust in the therapeutic alliance; in addition, the PPCE could provide false reassurance when patients feel better before receiving treatment for their underlying condition [7, 71]. Therefore, further study of therapeutic POCUS must be paired with diligent exploration of its ethical implications. Ultimately, therapeutic POCUS has potential to improve patient wellbeing in the hospital and may facilitate broader use of POCUS as a diagnostic and treatment modality.

## Supporting information

**S1 Raw data.**
(XLSX)

**S1 Fig. Actual versus patient- and provider-estimated duration of POCUS encounter.** [1,2,3,4] [1]Nonparametric correlation test yielding Spearman $r_s$. [2]Linear regression with null hypothesis of slope = 1. [3]Darker circles indicate 2 overlapping data points. [4]Two pairs of data points were excluded from provider estimation analyses due to missing values.
(DOCX)

**S2 Fig. STAI-6 score[1] versus Δ Time (patient).** [2,3,4] [1]Lowest possible score = 6; highest possible score = 24. [2]Δ Time (patient) was calculated by subtracting actual duration of POCUS encounter from patient-estimated duration of POCUS encounter. [3]Linear regression with null hypothesis of slope = 1. [4]Darker circle indicates 2 overlapping data points.
(DOCX)

**S1 Table. Perceptions of study participants who completed post-encounter surveys (n = 27).** [1]Patient satisfaction survey utilized a 1–5 Likert scale, with 5 being 'very satisfied' and 1 being 'very dissatisfied.' Adapted from Howard et al., 2014 [3]. [2]Bedside ultrasound survey utilized a 1–5 Likert scale, with 5 being 'strongly agree' and 1 being 'strongly disagree.' [3]The responses to these survey questions were averaged to calculate the "magnitude of PPCE" variable. [4]Provider competence survey utilized a 1–5 Likert scale, with 5 being the highest and 1 being the lowest. Adapted from Claret et al., 2016 [4]. [5]Provider empathy survey required replacing "Dr. X" with the provider's name and utilized a 1–7 Likert scale, with 7 being 'strongly agree' and 1 being 'strongly disagree.' Adapted from Hojat et al., 2011 [12]. The responses to these survey questions were averaged to calculate the "provider empathy" variable.
(DOCX)

**S2 Table. Additional variables that did not correlate significantly with magnitude of PPCE.** [1]Spearman $r_s$ was calculated for all variables except patient's preferred language, for which point-biserial $r_{pb}$ was calculated. [2]Two pairs of data points was excluded from this analysis due to missing provider ratings. [3]One pair of data points was excluded from these analyses due to missing observer ratings. [4]Patient's preferred language was defined dichotomously as "English" or "Non-English" for statistical analysis.
(DOCX)

## Acknowledgments

The authors thank Dr. Tyrel Starks for critical review and comments during manuscript drafting; the inpatient internal medicine POCUS providers at NYP/WC for facilitating our patient recruitment and coordinating with our team; and the Cornell Center for Social Sciences for assistance with NVivo software. We are grateful to all the patients who participated in this study for sharing their time and perspectives.

## Author Contributions

**Conceptualization:** Tanping Wong, David A. Scales.

**Data curation:** Evan A. Balmuth, Danny Luan.

**Formal analysis:** Evan A. Balmuth, Danny Luan.

**Funding acquisition:** Tanping Wong, David A. Scales.

**Investigation:** Evan A. Balmuth, Danny Luan.

**Methodology:** Deanna Jannat-Khah, Arthur Evans, Tanping Wong, David A. Scales.

**Project administration:** Evan A. Balmuth, Danny Luan.

**Supervision:** David A. Scales.

**Writing – original draft:** Evan A. Balmuth, Danny Luan.

**Writing – review & editing:** Evan A. Balmuth, Danny Luan, Deanna Jannat-Khah, Arthur Evans, Tanping Wong, David A. Scales.

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
