## [Decision Letter · Decision Letter 0]

16 Jan 2024

PONE-D-23-41071Point-of-care ultrasound (POCUS): Assessing patient satisfaction and socioemotional benefits in the hospital settingPLOS ONE

Dear Dr. Balmuth,

Thank you for submitting your manuscript to PLOS ONE. After careful consideration, we feel that it has merit but does not fully meet PLOS ONE’s publication criteria as it currently stands. Therefore, we invite you to submit a revised version of the manuscript that addresses the points raised during the review process.

ACADEMIC EDITOR: This an interesting paper addressing an important area around POCUS in the hospital setting. Please attend to the reviewer comments carefully. In addition to the reviewer comments, please indicate in your discussion how your findings support or add to what is already known in the use if POCUS in patient care.

We look forward to receiving your revised manuscript.

Kind regards,

Aloysius Gonzaga Mubuuke

Academic Editor

PLOS ONE

Journal Requirements:

2. In the online submission form, you indicated that "The data presented in this study are available upon reasonable request to the corresponding author. The data are not publicly available due to sensitivity of the patients’ clinical information."

Reviewers' comments:

Reviewer's Responses to Questions

**Comments to the Author**

1. Is the manuscript technically sound, and do the data support the conclusions?

Reviewer #1: Yes

Reviewer #2: Yes

2. Has the statistical analysis been performed appropriately and rigorously? 

Reviewer #1: Yes

Reviewer #2: Yes

3. Have the authors made all data underlying the findings in their manuscript fully available?

Reviewer #1: No

Reviewer #2: Yes

4. Is the manuscript presented in an intelligible fashion and written in standard English?

Reviewer #1: Yes

Reviewer #2: Yes

5. Review Comments to the Author

Reviewer #1: Very interesting study on the patient satisfaction of POCUS; well written and easy to understand.

I have just one general comment:

In the limitations i think an added source of bias, might be from the providers. The providers, knowing that they are part of a study and being observed, may act more friendly and more in line with what they perceive to be good bedside manner. How did the researchers mitigate for this in the study?

Reviewer #2: Thank you for an interesting read on an essential topic in the space of the emerging new technology POCUS in the inpatient setting.

Please correct the following: Page 19 line 378 should read 'provider' instead of 'prover'.

Please specify and add some detail which specific POCUS devices were used and applied during the study as there are now also smaller and portable devices used globally.

6. PLOS authors have the option to publish the peer review history of their article (what does this mean?). If published, this will include your full peer review and any attached files.

Reviewer #1: No

Reviewer #2: **Yes: **Ulrich Steinwandel

---

## [Author Response · Author response to Decision Letter 0]

23 Jan 2024

To the Editors,

Thank you very much for reviewing our article, “Point-of-care ultrasound (POCUS): Assessing patient satisfaction and socioemotional benefits in the hospital setting.” Your insightful comments, and those of the reviewers, have pointed us toward important changes that have improved this article’s impact. Based on your comments, we have made several revisions throughout the paper. Having worked carefully to address the reviewers’ comments, we feel the manuscript is greatly improved, and we are grateful for the opportunity to resubmit our revised manuscript. 

Below we provide point-by-point responses to each of the reviewers’ concerns (note that all page numbers refer to the revised, unmarked version of the manuscript):

Academic Editor & Journal Requirements

1. This an interesting paper addressing an important area around POCUS in the hospital setting. Please attend to the reviewer comments carefully.

Thank you very much for considering our work, and for your insightful comments.

2. In addition to the reviewer comments, please indicate in your discussion how your findings support or add to what is already known in the use if POCUS in patient care.

Thank you for this important suggestion. In our discussion, we now indicate in the following instances how our findings support or add to what is already known in the use of POCUS in patient care:

- Page 20, lines 386–388: we indicate that our finding of patients feeling generally very satisfied with POCUS supports similar findings in prior publications across ED and outpatient settings. We have added a sentence to clarify that our study is, to our knowledge, the first to assess patient satisfaction with POCUS in an inpatient internal medicine setting.

- Page 23, lines 445–448: we indicate that our findings of provider reassurance through image interpretation promotes patient satisfaction supports prior work showing similar results in the setting of fetal ultrasound. We have added a sentence to clarify that our study is, to our knowledge, the first to identify a role for provider reassurance in promoting patient satisfaction with POCUS encounters.

- Page 26, lines 519–520: we indicate that, through this study, we provide the first characterization of the POCUS positive care effect and its facilitating factors in the inpatient hospital setting, implying an important addition to the POCUS literature.

- Page 26, line 520 to page 527, line 530 & lines 537–539: we indicate that our findings may lay the groundwork for establishing a “therapeutic POCUS” approach which can improve patient wellbeing and facilitate broader use of POCUS as a diagnostic and treatment modality, implying an important addition to the POCUS literature.

We have made formatting changes to ensure our manuscript now meets PLOS ONE’s style requirements to the best of our knowledge.

4. In the online submission form, you indicated that "The data presented in this study are available upon reasonable request to the corresponding author. The data are not publicly available due to sensitivity of the patients’ clinical information." … If your data cannot be made publicly available for ethical or legal reasons (e.g., public availability would compromise patient privacy), please explain your reasons on resubmission and your exemption request will be escalated for approval.

Thank you for requesting clarification on this point. In our new submission, we now include a file with the deidentified raw data that was used in our quantitative analyses. To ensure maximal patient privacy, we provide an age range for each participant instead of their exact age.

We now include our full ethics statement in the ‘Methods’ section of our manuscript file, which can be found from page 5, lines 112–114 to page 6, lines 115–118. This statement also includes explanations of our verbal consent procedures and waiver of informed consent.

6. Please review your reference list to ensure that it is complete and correct.

We have reviewed our reference list and ensured that it is complete, correct, and adheres to PLOS ONE’s style requirements to the best of our knowledge.

Reviewer #1

7. Very interesting study on the patient satisfaction of POCUS; well written and easy to understand.

Thank you very much for reviewing our work, and for your input and recommendations.

8. I have just one general comment:

In the limitations i think an added source of bias, might be from the providers. The providers, knowing that they are part of a study and being observed, may act more friendly and more in line with what they perceive to be good bedside manner. How did the researchers mitigate for this in the study?

Thank you for highlighting the important limitation of potential observer bias. In our methods section (page 8, lines 136–139), we describe the elements of our protocol intended to mitigate observer bias: instructing providers to perform POCUS as they normally would for routine care, ignoring observers in the room, and avoiding references to the research study. We have now added an additional sentence in the limitations section (page 26, lines 505–509) to further clarify the steps we took to mitigate observer bias, as well as suggest how this could be further mitigated in future studies.

Reviewer #2

9. Thank you for an interesting read on an essential topic in the space of the emerging new technology POCUS in the inpatient setting.

Thank you very much for reviewing our work, and for all insights and recommendations.

10. Please correct the following: Page 19 line 378 should read 'provider' instead of 'prover'.

Thank you for pointing us to this typo. We have implemented the suggested correction, which can now be found on page 20, line 371.

11. Please specify and add some detail which specific POCUS devices were used and applied during the study as there are now also smaller and portable devices used globally.

Thank you for pointing us to this omission; we agree that the specific POCUS device used is an important consideration. The GE Venue GO Ultrasound System was used in this study’s POCUS encounters; we now specify this in the methods section (page 8, lines 141–142).

On behalf of the entire research team, thank you again for considering our work for publication in PLOS ONE.

Sincerely,

Evan Balmuth, BS

MD Candidate 

Weill Cornell Medical College

---

## [Editor Report · Decision Letter 1]

30 Jan 2024

Point-of-care ultrasound (POCUS): Assessing patient satisfaction and socioemotional benefits in the hospital setting

PONE-D-23-41071R1

Dear Dr. Balmuth,

We’re pleased to inform you that your manuscript has been judged scientifically suitable for publication and will be formally accepted for publication once it meets all outstanding technical requirements.

Kind regards,

Aloysius Gonzaga Mubuuke

Academic Editor

PLOS ONE
---

## [Editor Report · Acceptance letter]

9 Feb 2024

PONE-D-23-41071R1 

PLOS ONE

Dear Dr. Balmuth, 

I'm pleased to inform you that your manuscript has been deemed suitable for publication in PLOS ONE. Congratulations! Your manuscript is now being handed over to our production team.

Kind regards, 

on behalf of

Dr. Aloysius Gonzaga Mubuuke 

Academic Editor

PLOS ONE